# Role of Integrins in Sperm Activation and Fertilization

**DOI:** 10.3390/ijms222111809

**Published:** 2021-10-30

**Authors:** Veronika Merc, Michaela Frolikova, Katerina Komrskova

**Affiliations:** 1Laboratory of Reproductive Biology, Institute of Biotechnology of the Czech Academy of Sciences, BIOCEV, Prumyslova 595, 252 50 Vestec, Czech Republic; veronika.merc@ibt.cas.cz (V.M.); michaela.frolikova@ibt.cas.cz (M.F.); 2Department of Zoology, Faculty of Science, Charles University, Vinicna 7, 128 44 Prague, Czech Republic

**Keywords:** integrins, sperm, sperm activation, oocyte, fusion, reproduction

## Abstract

In mammals, integrins are heterodimeric transmembrane glycoproteins that represent a large group of cell adhesion receptors involved in cell–cell, cell–extracellular matrix, and cell–pathogen interactions. Integrin receptors are an important part of signalization pathways and have an ability to transmit signals into and out of cells and participate in cell activation. In addition to somatic cells, integrins have also been detected on germ cells and are known to play a crucial role in complex gamete-specific physiological events, resulting in sperm-oocyte fusion. The main aim of this review is to summarize the current knowledge on integrins in reproduction and deliver novel perspectives and graphical interpretations presenting integrin subunits localization and their dynamic relocation during sperm maturation in comparison to the oocyte. A significant part of this review is devoted to discussing the existing view of the role of integrins during sperm migration through the female reproductive tract; oviductal reservoir formation; sperm maturation processes ensuing capacitation and the acrosome reaction, and their direct and indirect involvement in gamete membrane adhesion and fusion leading to fertilization.

## 1. Introduction

Mammalian reproduction is a complex physiological process that at the molecular level involves a series of events preceding successful sperm-oocyte fusion. This process is facilitated by mutual interactions of many molecules developing wide protein networks in membranes of both germ cells. The main participants of these complex protein networks are transmembrane tetraspanin receptors in cooperation with adhesion molecules called integrins. The integrins are heterodimeric transmembrane glycoproteins that represent a large group of cell adhesion receptors involved in cell–cell, cell–extracellular matrix, and cell–pathogen interactions [1]. These receptors have the ability to transmit signals in both directions, into and out of a cell, and are involved in a wide range of physiological processes such as: immune response, lymphocyte homing, platelet aggregation, in wound healing, cell differentiation, migration, proliferation, and even in cell survival [2,3]. To date, 18 α and 8 β subunits have been detected in mammalian cells, which form 24 different known heterodimer combinations [4,5]. The particular combination of subunits defines ligand binding specificity of individual integrin heterodimers. While some integrin heterodimers interact with only a single ligand others recognize several distinct proteins [6]. In addition to somatic cells, integrins have also been detected on both male and female germ cells and are known to play an important role in multiple levels during the complex event of mammalian reproduction. The present review provides a comprehensive overview of current knowledge of the role of integrins in gamete physiology, maturation, activation, communication, interaction, and fusion.

## 2. Integrins in Mammalian Gametes

Mammalian fertilization is based on cell–cell and cell–extracellular matrix interactions, and integrins, as key adhesive receptors of cells, are significantly involved in this process. The role of integrins in fertilization was investigated during the 1990s and most research was carried out on the study of their expression in oocytes. It was shown that the use of function-blocking monoclonal antibodies against a selected integrin during in vitro fertilization, reduced the ability of sperm to bind or fuse with an oocyte (reviewed in [7]). However, the experiments with integrin knock-out mice did not confirm the essential role of integrin in gamete membrane fusion. It is significant to note that much less attention was paid to researching the role of integrins on sperm. To date, 9 α and 7 β subunits have been described on oocytes, but only 5 α and 3 β subunits have been detected on sperm and are summarized in Table 1. 

These subunits create heterodimers, which localize differently in individual compartments of the sperm head (Figure 1). The integrins are part of a wide protein network called the tetraspanin web located in the oocyte plasma membrane (PM) called the oolemma. Within the tetraspanin web the integrins interact and cooperate with tetraspanins such as CD9, CD81, and CD151, as well as other receptors. Some of the integrins and tetraspanins identified on the oocyte have also been found on sperm. For this reason, it is likely that similar protein networks to those on the oocyte exist on sperm, where they participate in sperm maturation, activation, and interaction with the surrounding epithelium and ultimately with the oocyte.

### 2.1. Integrins as a Part of the Oviductal Reservoir of Sperm

During sexual intercourse in mammals, sperm enters into the female reproductive tract (the vagina or uterus, depending on the species [19,20]), and migrates into the site of fertilization, to fuse with the oocyte. During this journey, only a part of the ejaculated sperm (that without morphological defects) reaches the fallopian tube [20,21].

During the migration from uterus to the oviduct, particularly to its proximal part called *ampulla*, sperm have to overcome crucial barriers in the female reproductive tract called uterotubal junctions (UTJ). The UTJ serve as an active selective filter for sperm transport [22] and it is known that various proteins of otherwise morphologically intact and normally motile sperm play a role in this selection [23]. Experiments with mice lacking genes for ADAM proteins, especially ADAM3 [24,25], point to the inability of such spermatozoa to pass through the UTJ. In this case, the integrins expressed on UTJ epithelial cells could potentially be binding partners for sperm proteins of the ADAM family [26]. The failure of the ability of sperm to bind to the integrins of the oviductal epithelium, leads to an inability of sperm to migrate through the UTJ, however the specific mechanism of this process is not yet known [27]. Since such sperm are also reported to have a reduced ability to bind to *zona pellucida* of the oocyte, it is possible that a similar mechanism applies to both the above-mentioned interactions [26]. The theory, that integrins may play a role in the transfer of sperm through the UTJ, is supported by the fact that sperm with missing genes for other proteins, such as calmegin, lack the ability to pass through the UTJ [28]. Calmegin serves as chaperone protein and its failure leads to incorrect folding of the protein in the endoplasmic reticulum. Mice without calmegin produce sperm without ADAM2 in their PM [29] and these sperm also are not able to overcome the UTJ [24].

Those sperm, which pass through the UTJ, bind via sperm head receptors to the epithelial cells in the narrowed section of the fallopian tube (*isthmus*), creating an oviductal reservoir [23,30], however the molecular mechanism is not currently clear. It has been shown that protein binding may vary from species to species, and at the same time may be mediated by complex protein interactions of different molecules in a particular species (e.g., [31]). Many studies have suggested that binding is mediated by glycans [32,33], and, in some cases, by fucose [34]. It also appears that integrins may be involved in the binding of sperm to oviductal epithelial cells. One possible mechanism is the binding of fibronectin, which is located on the apical part of the ciliated cells of the human oviduct, and integrin α5β1, which was detected on the surface of sperm [35]. Fibronectin binds to integrins through its RGD domain. Integrins have also been detected on the surface of oviductal ciliated cells, e.g., αV, β3, and β1 [27] and therefore, may also be involved in this interaction. These oviduct-specific integrins could potentially bind to sperm-localized fibronectin [31].

In addition to integrin interactions, other molecules of epididymal origin have been detected in sperm, which are probably involved in sperm transport through the female reproductive tract; examples are, β-defensin [31] or lectin-like molecules as Binder of SPerm (BSP) proteins [36]. While the β-defensin binding partner on oviductal epithelial cells is unknown, BSPs bind to annexins. Chaperone proteins in the oviduct are also proposed to be involved in the interaction with sperm, but their exact function is not yet known [37]. Research suggests that the binding of sperm to oviductal epithelial cells prolongs their life by regulating the course of capacitation and affecting sperm motility thus ensuring synchronization of sperm maturation with ovulation [38]. It is believed that this binding process also enables a selection of sperm with the best morphology, as well as enabling the gradual release of sperm capable of fertilization. The binding process increases the chances of successful fertilization while also reducing the likelihood of polyspermy [39,40]. 

Interestingly, the rate of sperm release increases during ovulation [41]. Due to the interaction of sperm with the components of oviductal fluid, capacitation is initiated and the ability of sperm to bind to epithelial cells in the fallopian tube is gradually lost. It was discovered in cattle [35,42] that an increase of fibronectin levels in an oviductal fluid during the pre-ovulatory period may promote a sperm release from the oviductal reservoir through the interaction with the sperm surface fibronectin receptor α5β1 integrin.

### 2.2. Integrins in the Interaction of Oviductal Fluid with Sperm

Integrins together with tetraspanins (e.g., CD9 and CD81) play a key role in the interaction of sperm with components of oviductal fluid. Reproductive fluids in both male and female genital tracts are known to contain extracellular microvesicles and exosomes. These are membrane-coated vesicles used for transferring the required proteins onto the sperm surface during maturation in the epididymis and subsequently in the female reproductive tract [43]. Upon binding to the oviductal reservoir, sperm comes into contact with oviductal fluid vesicles, so-called oviductosomes [44], which contain large numbers of proteins [45].

Unlike somatic cells, sperm are not capable of endocytosis, where the transfer of proteins, peptides, lipids, DNA fragments, and RNA molecules between extracellular vesicles and the recipient cell occurs by endocytosis through membrane fusion [46]. Therefore, fusion is the primary mechanism for protein transfer between sperm and microvesicles [47]. The crucial role of integrins in this fusion process has been confirmed by the research of Al Dossary et al. [48], who studied the presence, localization, and role of the CD9 molecule and the αV subunit of integrin αVβ3 in a mouse model. Both molecules were detected in oviductosomes, and the regions of sperm involved in the primary sites of gamete membrane fusion (the head and neck). The study showed that the fusion of the oviductosome with the sperm membrane can be inhibited in the presence of ligands (fibronectin and vitronectin) binding to the integrins α5β1 and αVβ3, by adding the Arg-Gly-Asp peptide chain (RGD motif) or antibodies against the αV subunit. This confirms that integrins play a crucial role in this process. According to the model of Al-Dossary et al. [48], the CD9 proteins form a fusion site on both membranes of oviductosomes and sperm (Figure 2), where adhesive proteins such as integrins α5β1 and αVβ3 are also present. The respective ligands for the described integrins, fibronectin and vitronectin, bind to their activated receptors, bringing the membrane of sperm and oviductosome to a distance of less than 0.5 nm [49], which ensures electrostatic repulsion of polar lipid heads. This results in the opening of the plasma membrane outer layer, leading to the formation of hydrophilic fusion pores between the sperm and the oviductosome and thus, fusion. In addition to mouse sperm, αV integrin was detected on human sperm [17] where it is localized in the PM covering the acrosomal cap (AC) and the inner acrosomal membrane (IAM). This may suggest that a similar mechanism as the one proposed by Al-Dossary et al. [48] for mouse, may play a role in the fusion of the human sperm membranes with oviductosomes in the reproductive tract of women.

The described sperm-oviductosomes membrane interaction is important, for example, for the transport of the membrane protein Ca^2+^ ATPase (PMCA4), which serves to transfer Ca^2+^ across the PM of sperm [50]. Sufficient PMCA4 is important for maintaining sperm viability during capacitation, hyperactivation, and the acrosome reaction and all these processes require elevated Ca^2+^ levels [51]. Furthermore, transmembrane and membrane proteins can be transported to the sperm surface via oviductosomes during these final maturation processes [48]. Interestingly, Ferraz et al. [45] observed higher success rates of fertilization using in vitro experiments on a cat model when sperm were incubated with oviductosomes compared to the samples where oviductosomes were not present. At the same time, it is quite probable that the same or a similar mechanism, based on integrins and tetraspanins, also applies to the interaction of sperm with extracellular vesicles in the earlier stages of maturation, e.g., in the male reproductive tract.

### 2.3. The Role of Integrins in Membrane Reorganization and Stability

During the acrosome reaction, a significant reorganization of sperm membranes occurs, involving the fusion of the outer acrosomal membrane (OAM) with the PM that is facilitated by rapid depolymerization of actin fibers [52] and leads to exocytosis of the acrosome [53]. The released content disrupts *zona pellucida* and allows the sperm to enter the perivitelline space, and thus come into direct contact with the oolemma of the oocyte. At the same time, the IAM, with its protein content, reaches the surface of the sperm head [54,55]. During the acrosome reaction, proteins from the apical acrosome (AA), such as Izumo1 [56], CD9, CD81, CD46 [10,57], and integrins [10,12] relocate to other compartments of the sperm head, mainly to the primary fusion site of sperm formed by IAM and residual PM called the equatorial segment (ES). These relocated proteins later participate in gamete membrane interaction and fusion. The dynamic relocation of integrins during the final stages of sperm maturation suggests their possible function in processes following the acrosome reaction. Feasibly, protein relocation occurs by transport via hybrid vesicles, which are released during the fusion of the PM and OAM followed by fusing with the remaining PM covering the sperm [58]. It is known that, in addition to integrins, these vesicles also contain, for example, the CD46 protein [10].

Actin networks play a key role in signaling pathways and processes during the acrosome reaction. An open question remains regarding which specific proteins in addition to actin may be involved in the reorganization and stabilization of this protein network. Integrins are suspected to play a crucial part due to their ability to interact directly or indirectly with the actin cytoskeleton (Figure 3) and control and regulate its rearrangement [59]. The indirect interactions between actin and integrins are achieved via their binding partners within the membrane protein network such as CD46 [60], CD81 [61], and CD9 [62]. Based on recent research, which extended our knowledge of the existence of the tetraspanin network in sperm [11,12], we could argue that the same mechanism also takes place on sperm. It has been shown that the CD46 transmembrane protein represents a binding partner of β1 integrins in sperm and both these proteins jointly relocate during the acrosome reaction [10]. In addition, CD46 is responsible for stabilizing the acrosomal membrane and thus the entire acrosome via the actin cytoskeleton. This acrosome stability is achieved either via direct binding to actin filaments in the presence of the ERM (ezrin, radixin, moesin) protein, or indirectly via interaction with β1 integrins. Interestingly, it has been shown that mouse sperm depleted in the CD46 protein [63] have an increased rate of spontaneous acrosome reaction that is initiated in the absence of *zona pellucida*. This observation correlated with the research done on field mice (genus *Apodemus*) in which sperm displayed a high rate of spontaneous acrosome reaction due to the absence of the CD46 protein [64,65,66]. The above supports the theory that the CD46 protein participates in acrosome stability, possibly in conjunction with β1 integrins.

In addition to β1 integrins, β4 integrins, which have been detected in both human [8] and mouse sperm [11], are possibly involved in acrosome stability and the acrosome reaction. Frolikova et al. [11] detected β4 integrin subunits in the PM overlying the apical part of the acrosome and in the ES of mouse sperm. The cytoplasmic tail of β4 integrin is significantly longer than other β integrin subunits and has the ability to be involved in cytoskeletal reorganization. It affects, for example, the Rac1 protein (Ras related C3 botulinum toxin substrate 1 protein) and its activation [67], which plays an important role in capacitation and the acrosome reaction, especially in the remodeling of actin in the apical part of the acrosome [68]. Unlike other integrins, β4 binds not only to actin and tubulin but also to intermediate filaments, such as keratin 5 via its counterpart plectin, which surrounds the sperm nucleus [68]. This binding ability is due to the structural differences of the cytoplasmic domain of β4 (Figure 3) and may contribute to the mechanical and structural stability of a cell. In a similar way to β1 integrins, it is possible that β4 integrins interact with tetraspanins. Jankovicova et al. [12] presented in a recent study, an expression of CD151 protein in the sperm ES and confirmed its interaction with the α6 integrin subunit. This protein assembly should result in the formation of a stabilizing protein complex at this primary fusogenic site during the acrosome reaction.

### 2.4. Integrins in the Organization of Protein Complexes

Tetraspanins are the main binding partners of integrins in gametes. Several tetraspanins have been found in sperm (Figure 1), mainly, CD9 [57,69,70,71], CD81 [57,72], and CD151 [12]. They are proteins with four transmembrane domains and two extracellular regions, and their main role is to regulate the arrangement of membrane protein complexes (such as receptors, signaling proteins, and fusogens). Tetraspanins participate in the reorganization of membranes, relocation of proteins, stabilization of individual sperm compartments, and can modulate the function of other members of protein networks, mainly through association with integrins such as α3β1 and α6β1 [73] or αVβ3 [74]. These bonds can be either direct (CD151) or indirect through other tetraspanins (CD9 and CD81).

CD9 is an essential protein in the oocyte. Anti-CD9 antibodies significantly reduce fertilization in vitro [75] and the depletion of *Cd9* gene results in almost complete sterility of female mice, while males remain fertile [73,76]. In oocytes, the main role of CD9 proteins is not only an interaction with other proteins (e.g., integrins) within the oolemma, called *cis*-interactions (lateral bonds between proteins or subunits in the membrane of one cell), but also protein interactions between the oolemma and sperm membrane, called *trans*-interactions (antiparallel interactions between proteins or subunits between membranes of two cells to mediate cell-cell adhesion) (Figure 4). Moreover, CD9 is responsible for the formation of egg *microvilli* [77] and its crystal structure analysis suggests that this molecule is localized in *microvilli* high curvature regions [78]. CD9 is probably involved in enhancing adhesion between gametes implying the *trans*-protein interaction [57]. In addition, Miyado et al. [79] suggested that CD9 plays a major role in the formation of microvesicles that are released from the oocyte into the environment ultimately before fertilization. The transport of CD9 from the CD9-positive oocytes to sperm via exosomes, allowed these sperm to bind to CD9-deficient oocytes [79,80]. However, these results could also be interpreted with later discovery of CD9 expression in sperm [57,69]. It is also feasible that other proteins such as integrins could be exchanged between gametes via exosomes [18].

In addition to CD9, the role of the CD81 tetraspanin was addressed by the *Cd81^−/−^* mouse model. The *Cd81*-depleted mice did not show a significant reduction in fertility compared to *Cd9*^−/−^, however double CD9^−/−^/CD81^−/−^ mouse model resulted in total sterility of females [76]. Importantly, the research shows that the role of tetraspanins may be species specific, as similarly described for integrins, and it is relevant that anti-CD81 antibodies significantly reduced fertilization in mice [76], whereas they had no effect on human oocyte fertilization success [81]. On the other hand, anti-CD151 antibodies blocked sperm-oocyte membrane fusion mainly in human [81].

### 2.5. The Role of Integrins in Gamete Fusion

After sperm penetrate *zona pellucida* and thus enter the perivitelline space, they bind to the oolemma and one of them subsequently fuses with it. To date, many candidate proteins have been proposed, including integrins, that may play a key role during gamete binding and in membrane-specific fusion regions. The sperm head membrane is divided into specific domains destined to come into first contact with the oolemma, and their specific functions are reflected in their unique protein profile [82]. In contrast, the oolemma, is compartmented into *microvilli*, where sperm binding occurs [83,84], and areas that are not specialized for sperm primary binding such as, oolemma between individual *microvilli* and in some species a distinct *microvilli*-free region overlaying the female nucleus. Gamete fusion is a very complex process, and it involves a number of interactions of protein networks located in the membranes of both gametes. Based on experimental work using monoclonal antibodies, in vitro fertilization, or knock-out models, important components of these protein networks were identified. Additionally, certain candidate proteins were ruled out (e.g., fertilin or CRISP1 [85]). Based on current research, Juno and CD9 on oocytes; and Izumo1, SPACA6, TMEM95, FIMP, SOF1, DCST1, and DCST2 on sperm (reviewed in [86]) are considered as fertilization-essential proteins in mammals [73,87,88,89,90,91,92,93,94]. 

Currently, the role of integrins in sperm-oocyte binding/fusion is still not fully understood. Initial hypotheses about the possible role of integrins in gamete adhesion/fusion stemmed from the discovery of their binding partners, proteins from the ADAM family in sperm [95,96]. Initially, fertilin, a heterodimer consisting of two subunits, fertilin α (ADAM1B) and fertilin β (ADAM2), were investigated. Experiments with antibodies against fertilin β and the *fertilin β* knock-outs resulted in a significant reduction in sperm adhesion and fusion to the oocyte [24,96,97,98]. Subsequently, ADAM proteins have also been shown to be engaged in sperm migration into the oviduct and in binding to the *zona pellucida*. Most of the ADAM proteins contain a disintegrin (integrin ligand-like) domain through which they can bind to integrins, demonstrated in somatic cells [95]. The presence of integrins was soon confirmed in the oocyte [99,100,101] and the interaction of integrins and their receptors, ADAM proteins on sperm, was suggested as a mechanism necessary for membrane adhesion and fusion. Even though the integrin α6β1 was proposed as a main binding partner for fertilin β [102], experiments using antibodies did not provide a consistent outcome. Almeida et al. [103] showed that antibodies against the α6 integrin subunit significantly inhibited sperm binding to *zona pellucida*-free oocytes, while Evans et al. [97] did not observe such an effect. It should be noted that in both cases a different method of removing *zona pellucida* was used; which could have affected the results because methodology can significantly change the oocyte environment and distribution of integrins and tetraspanins. Further, the inhibitory effect of anti-α6 integrin antibody was not observed by Evans [104], while Miller et al. [105], using oocytes with *zona pellucida* and *cumulus oophorus* cells attached concluded that the α6 integrin subunit was not involved in gamete interaction in mice. Barraud-Lange et al. [16] pointed out, however, that the methodology for detecting fertilized oocytes used by Evans [104] could lead to false positive results. In addition, Miller et al. [105] performed in vitro fertilization in small volumes, which could result in reduced fertilization in the control group affecting the overall assumptions. Interestingly, all four studies showed that antibodies to β1 integrins prevented fusion, suggesting that in oocytes, β1 subunits may bind to other α subunits, and these heterodimers could engage in gamete adhesion and fusion [97,103,104,105]. Experiments using antibodies suggested the possible role of various oocyte integrin subunits also in other mammalian species, such as in cattle [106,107,108,109]. Interestingly, integrins may also influence the morphology and quality of the oocytes. It was shown that an increase of integrin protein level positively corelates with oocyte quality and morphology prior to fertilization in pigs [110].

Later, the knock-out models shed new light on the role of integrins during gamete binding/fusion. Due to the fact that integrins are widely distributed in various cell types, and their systemic depletion often leads to premature lethality or developmental defects [5], the methodology must have been adapted accordingly (collection of ovaries after birth, or using a conditional gene knock-out line strategy). Using this approach, the sterility was not confirmed in *β3 integrins* [111], or *α3*/*α6 integrins* [105,112] knock-out mice. Further, He et al. [112] using the *β1 integrin* knock-out line found that oocytes were capable of binding and fusion with the sperm. Inhibition did not occur even when such oocytes were incubated with antibodies against β3 and αV integrin subunits. Based on all available knowledge, it was concluded [112] that none of the integrins expressed on oocytes were essential for sperm–oocyte binding and fusion in mouse, which was further supported by Florman and Fissore [113]. Interestingly, it was later shown that partial *α9 integrin* knock-out (resulting in about 50% of protein expression) causes a noticeable reduction in fertilization capability in mice [114]. It is plausible that in the absence of one integrin subunit, another takes over its place. A presence of the α9β7 integrin heterodimer on human lymphoblastic B cells of the RPMI 8866 line, which do not naturally contain the β1 subunit was detected and its ability to bind to the ADAM2 molecule was shown [115]. Further, theoretical structure modelling research has shown that other yet undiscovered combinations of integrin subunits could be predicted [116]. It is important to note that gamete adhesion and fusion is a species-specific process and while anti-α6 antibodies have a negligible effect in mice, they significantly inhibit fusion in human [81,117] and partial inhibition (up to 55%) in human gametes was further observed when the integrin subunits such as α2, α3, α5, αV, αM, β1, β2, and β3 were targeted [117].

Additionally, there was strong evidence of the existence of integrins in sperm [8,13,14], which was known at the time of anti-integrins antibody blocking and gene knock-out experiments performed with oocytes (described above). Surprisingly, none of the mentioned studies addressed any sperm specific role of integrins as possible contributors to fertilization outcome. In addition, in documented oocyte experiments, wild-type sperm were used for in vitro fertilization and wild-type males were used for mating and monitoring fertilization success in vivo. As subsequent experiments have shown, incubation of mouse sperm with anti-α6 [16], anti-αV, or anti-β3 [18] integrin antibodies, or bovine sperm with anti-β1 [118] or anti-β5 [119,120] integrin antibodies, reduced the ability of sperm to bind to the oocyte. In mice, this effect was more pronounced when both gametes were incubated with the antibodies prior to their interaction. It was suggested that under optimal conditions, integrins in both gametes are involved in the interaction, but their presence in either sperm or oocyte is sufficient for successful fertilization, and further, there may be various protein exchanges between gametes [18]. Castellano et al. [121] observed a significant linear correlation between the fertilization rate and the percentage of bovine sperm with detectable α5β1 integrin in the ES and post-acrosomal region (PAR) suggesting its role in gamete fusion.

Recently, Barraud-Lange et al. [122] demonstrated that the conditional knock-down of the sperm *Itgb1* gene in mice resulted in a dramatic decrease of the fusogenic ability of such sperm. In vitro, sperm accumulated in the perivitelline space (similarly to *Izumo1*^−/−^ or *SPACA6*^−/−^), independently whether oocytes were depleted of *Itgb1* gene or not, providing the evidence that sperm β1 integrin is crucially involved in the gamete adhesion/fusion process. Interestingly, individuals with knock-down *Itgb1* gene were fertile in vivo, probably as a result of the incomplete depletion of the *Itgb1* gene. The authors hypothesized that the sperm β1 integrin subunit might be involved in the binding of sperm to the oviductal epithelial cells and consequently only sperm with the β1 integrin subunit present, which are destined to reach the oocyte. In addition, the role of integrins on sperm during oocyte binding and fusion, is strongly supported by the fact that during the acrosome reaction, these proteins (observed in mouse, human, bovine, and porcine sperm) relocate into the ES [10,12,17,121,123]; in a similar way to other essential molecules (e.g., Izumo1, SPACA6) necessary for successful fusion.

The co-existence of integrin subunits on sperm and oocyte provides another possibility for sperm-oocyte interaction, in addition to the originally proposed binding between ADAM proteins and their integrin receptors on the oocyte. Integrins in gametes can also interact through ligands which they are able to bind to, such as fibronectin or vitronectin that were detected in sperm and result in the reduction of gamete binding and fusion [18]. At the same time, the interaction of integrins on sperm with so far unknown molecules on the oocyte is also possible (Figure 4) [87]. Integrins probably play an indirect role in gamete binding and fusion, as a possible secondary receptor for Izumo1, as a potential binding partner of SPACA6 protein on sperm or through interaction with tetraspanins. In addition, although it is likely that there are other complementary molecules that can complement integrin function, their reduced expression in gametes may be an evolutionary disadvantage, such as in case of lack of β1 integrins, which causes slower sperm binding [124].

#### 2.5.1. Integrins as a Potential Secondary Receptor of Izumo1

One of the essential proteins expressed in sperm is the membrane protein Izumo1, a member of the type I immunoglobulin superfamily with an extracellular immunoglobulin domain [91]. The binding receptor for Izumo1 on the oocyte is GPI-anchored protein Juno [88]. After recognition of the Juno by Izumo1 protein, there is expected to be a secondary binding of Izumo1 to another ligand, possibly a member of integrin family, another adhesion molecule, or a non-proteinaceous factor such as a phospholipid [125].

Izumo1 is detected on sperm only after an acrosome reaction during which it relocates to the ES [56]. The use of antibodies against Izumo1 prevented the fusion of sperm with oocyte, and at the same time *Izumo1*^−/−^ males were sterile, despite their normal behavior, development, sperm motility, migration through the female oviduct, acrosome reaction, and penetration through *zona pellucida*. These *Izumo1*^−/−^ sperm accumulated in the perivitelline space of the oocyte and were incapable of fusion with oolemma. Such phenotype could be overcome by intra-cytoplasmic sperm injection (ICSI), when further development in the absence of Izumo1 was not disrupted [91]. Similarly, in the case of Juno, females lacking this protein were shown to be sterile, despite the fact that after fertilization, Juno is shed from the oolemma, which suggests its secondary role in blocking of polyspermia [88].

#### 2.5.2. Integrins as a Potential Binding Partner of SPACA6

Integrins have also recently been proposed as a possible binding partner of the sperm acrosome membrane-associated protein 6 (SPACA6). Through integrins, SPACA6 can also interact and influence other molecules within a sperm membrane (Figure 4). However, this has not yet been experimentally confirmed, and it is therefore possible that SPACA6 binding to these molecules occurs via proteins other than integrins, or directly [87]. SPACA6 is a transmembrane immunoglobulin-like protein that is localized, as well as integrins, in the ES of sperm after the acrosome reaction [87,126]. Its key role in sperm fertilization ability was confirmed by experiments with antibodies applied to human gametes and by *SPACA6*^−/−^ in mice. The male phenotype in this case is very similar to that of *Izumo1^−/−^*, where sperm accumulate in the perivitelline space and are incapable of oolemma fusion [87]. The SPACA6 binding partner on the oocyte is not yet known.

SPACA6 and all other molecules (CD9, Juno on oocyte and Izumo1 on sperm), which are considered essential for successful fertilization, are more likely to play a role during sperm–oocyte binding than fusion, due to an absence of fusogenic domains known from, e.g., viral fusion systems [127,128]. The identification and gamete localization of the fusogen remains an open question. A drawback of experiments using genetically modified organisms is, pathological embryonic development and decreased early postnatal survival rate, including the occurrence of fatal prenatal defects or those which prevent an organism from reaching a reproductive age. On the other hand, a biochemical approach based on blocking the functions of a protein may not reveal a fusogen, if it is present in a low concentration or if it is temporarily activated [127]. Over the last three decades, several other molecules have been discovered in genomic and proteomic analyses e.g., SLLP1 protein on sperm and its partner SASB1 on oocyte; [129,130], and they are to a certain extent involved in mediating binding/fusion of sperm and oocyte. This only confirms the fact that in both gametes there are other molecules within multiprotein networks, which play an important role in fertilization through mutual bonds where integrins are expected to be involved.

## 3. Conclusions

This review delivers a comprehensive overview of existing literature on integrin specific expression and localization in mammalian gametes including known specie-specific differences. The function of integrins in sperm maturation and sperm transport through female reproductive epithelia is addressed including the relocation of integrins during the acrosome reaction, as summarized in Table 1. A discussion about the role of integrin during gamete membrane binding and fusion with respect to the sperm and oocyte specific interactive protein network is debated. Furthermore, the review presents novel diagrams summarizing the current state of knowledge which are produced for a wide-ranging scientific audience.

## Figures and Tables

**Figure 1 ijms-22-11809-f001:**
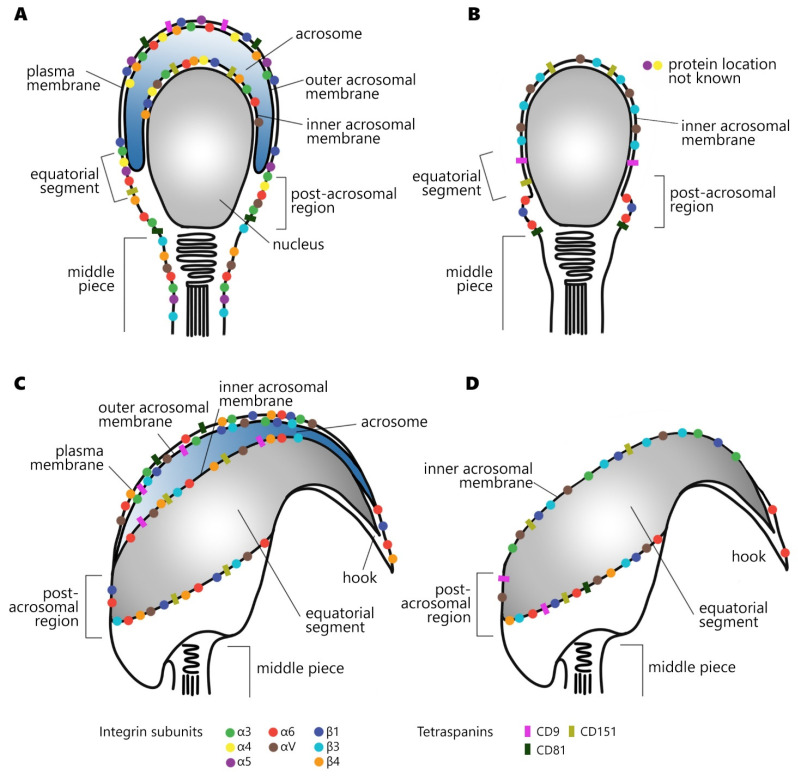
Localization of integrin subunits and tetraspanins in human (**A**,**B**), mouse (**C**,**D**), acrosome-intact (**A**,**C**), and acrosome-reacted sperm (**B**,**D**).

**Figure 2 ijms-22-11809-f002:**
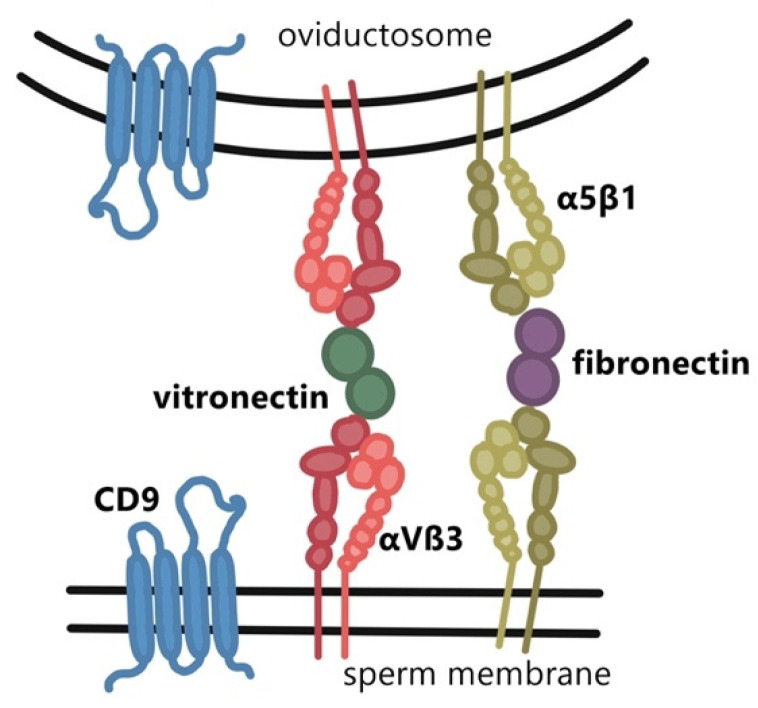
Interaction of sperm and oviductosome membranes via integrin heterodimers.

**Figure 3 ijms-22-11809-f003:**
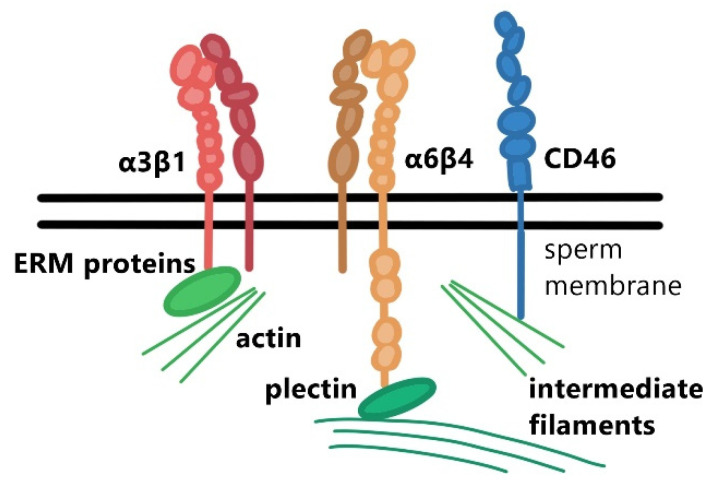
Schematic depiction of integrin heterodimers and the CD46 protein anchored in a sperm membrane facilitating interaction with the cytoskeleton via ERM proteins (ezrin, radixin, moesin) or plectin.

**Figure 4 ijms-22-11809-f004:**
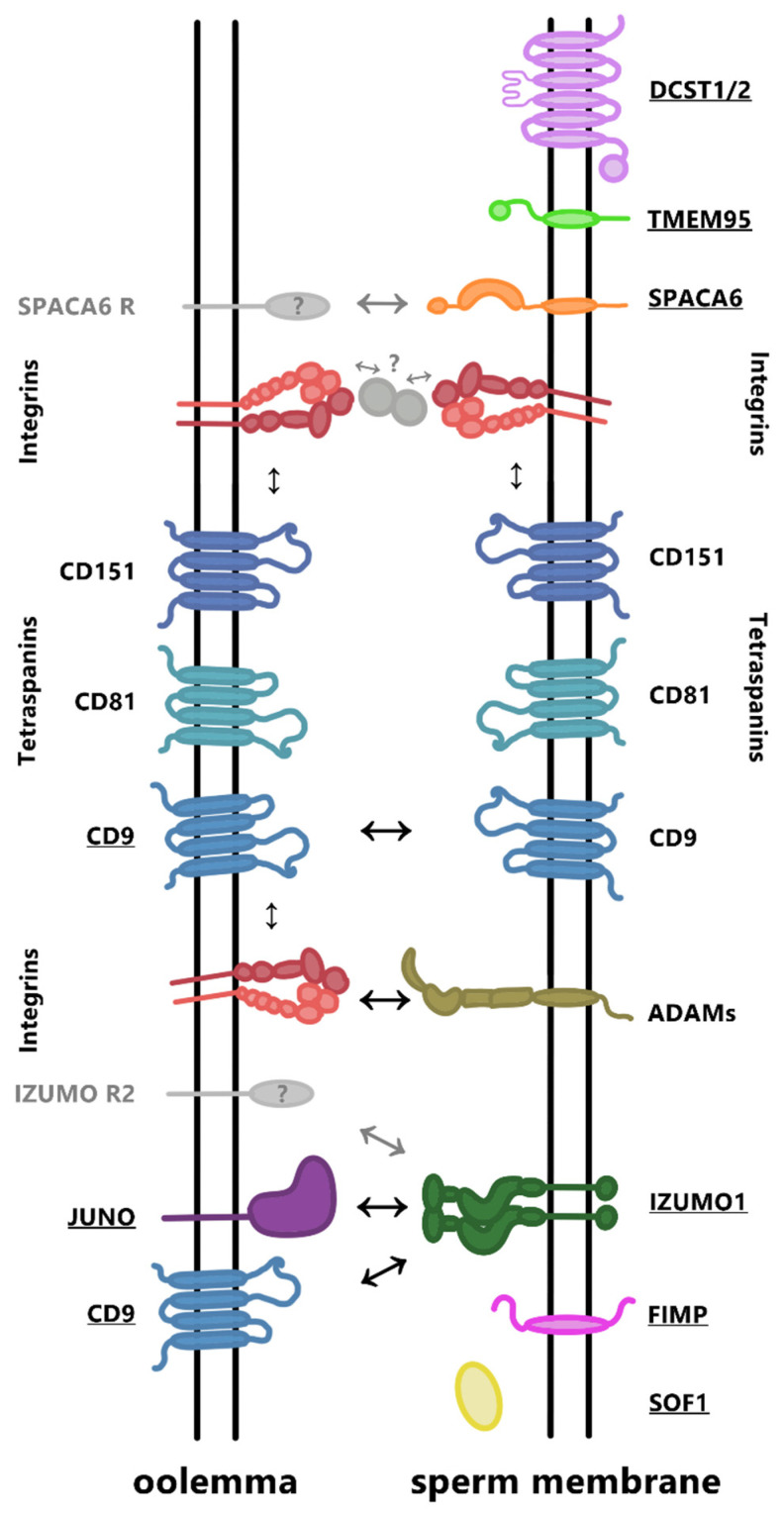
Hypothetical model of adhesion/fusion of mammalian gametes. The underlined molecules are considered essential in this process or yet unknown (?). Confirmed and possible interactions are depicted by black and grey arrows, respectively.

**Table 1 ijms-22-11809-t001:** Localization of integrin subunits on acrosome-intact and acrosome-reacted sperm in human and mouse. Abbreviations are as follows: (PM) plasma membrane; (OAM) outer acrosomal membrane, (IAM) inner acrosomal membrane, (AC) acrosomal cap, (AA) apical acrosome, (AH) apical hook, (PAR) post-acrosomal region, (ES) equatorial segment, (M) middle piece, and (*) protein localization not reported.

Subunit	Human Sperm		Mouse Sperm	
	Acrosome-Intact	Acrosome-Reacted	Acrosome-Intact	Acrosome-Reacted
α3	mainly AC (20% P + M) [8] ES [9]		OAM + PM [10] AC + OAM [11] AA [12]	AA [10]
α4	ES [9] mainly AC [8] * [13]	* [13]		
α5	ES [9] PM [14] ES + M (25% P) [8] * [13]	* [13]		
α6	ES [9] PAR + M (40% AC) [8] AC [15] * [13]	PAR [15] * [13]	ES + PAR [16] PM + AH + ES [10] AC + AH + ES [11] ES [12]	AH + ES [10] ES [12]
αV	IAM [14] PAR + M [8] ES + PM + OAM + IAM [17]	IAM [14] ES + PAR + IAM + M [17]	AC + ES [18] IAM + OAM + PM + AH [17]	AC + ES [18] PAR + IAM + M [17]
β1	ES [9] PM [14] AC [15]	PAR [15]	ES + P [16] OAM + AA + AH + PM [10] AC + OAM [11] AP [12]	ES + IAM + whole head [10]
β3	PAR + M [8]	IAM [14]	AC + ES [18]	AC + ES [18]
β4	AC (40% ES + M) [8]		AA + AH + ES [11] ES [12]	ES [12]

## Data Availability

Not applicable.

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
