# Peer review of "Role of Integrins in Sperm Activation and Fertilization"

_ijms, 2021, doi:10.3390/ijms222111809_

Round 1

Reviewer 1 Report

Dear Authors,

In the manuscript entitled “Role of integrins in sperm activation and fertilization”, you summarize the current knowledge on integrins in reproduction discussing on several aspects including their role in different phases of male sperm activation, the interaction with the oviductal fluid and during gamete fusion. This review discusses in a more deep way the topic described in some old reviews giving more emphasis to the interaction of sperm with oviductosome.  

The manuscript is sometimes difficult to read and the topic of integrins role in sperm activation and fertilization is investigated considering human and mice species albeit further data are present for bovine and swine species. The authors should improve the manuscript by adding finding on these animals species [see references Antosik P. et al., Veterinarni Medicina, 55, 2010 (4): 154–162; Gonçalves R.F. et al., Advances in Bioscience and Biotechnology, 2013, 4, 617-622; Velho A. et al., Andrologia 2019, 51(7):e13305; Castellano L. et al., Theriogenology 168 (2021) 66e74].

Author Response

Dear reviewer,

thank you very much for your valuable point to our manuscript, we addressed it and the information about the role of integrin in reproduction in bovine and porcine model was incorporated into the text. Please see the updated manuscript.

Reviewer 2 Report

The manuscript needs only few minor improvements prior being published.

Merc and the colleagues wrote a review on integrin roles in sperm activation and fertilization. The manuscript is interesting, well-structured with broad literature resources. I read it with great interest and found nearly no need for the improvement. Several rather minor points should be considered:

Minor comments:

  1. The localization of tetraspanins along with integrins in sperm cells would be highly interesting (Figure 1)
  2. Spelling the abbreviations in the Figure legends need to be done (e.g., ERM in Figure 3)

Author Response

Dear reviewer,

Thank you very much for valuable points to our manuscript, we addressed them. Please see the response below and updated manuscript.

Minor comments:

  1. The localization of tetraspanins along with integrins in sperm cells would be highly interesting (Figure 1)

The figure 1 was updated and it shows the localization of both integrins and tetraspanins. Please see updated figure.

  1. Spelling the abbreviations in the Figure legends need to be done (e.g., ERM in Figure 3)

The abbreviations in the Figure legends were explained. Please see figure legends.